# A New Flavanone from *Chromolaena tacotana* (Klatt) R. M. King and H. Rob, Promotes Apoptosis in Human Breast Cancer Cells by Downregulating Antiapoptotic Proteins

**DOI:** 10.3390/molecules28010058

**Published:** 2022-12-21

**Authors:** Gina Mendez-Callejas, Ruben Torrenegra, Diego Muñoz, Crispin Celis, Michael Roso, Jojhan Garzon, Ferney Beltran, Andres Cardenas

**Affiliations:** 1Grupo de Investigaciones Biomédicas y de Genética Humana Aplicada (GIBGA), Laboratorio de Biología Celular y Molecular, Facultad de Ciencias de la Salud, Universidad de Ciencias Aplicadas y Ambientales (U.D.C.A), Calle 222 # 55–37, Bogotá 111321, Colombia; 2Grupo de Investigación en Productos Naturales (PRONAUDCA), Laboratorio de Productos Naturales, Facultad de Ciencias, Universidad de Ciencias Aplicadas y Ambientales (U.D.C.A), Calle 222 # 55–37, Bogotá 111321, Colombia; 3Grupo de Investigación en Fitoquímica (GIFUJ), Departamento de Química, Facultad de Ciencias, Pontificia Universidad Javeriana, Cra. 7 # 40–62, Bogotá 111321, Colombia

**Keywords:** *Chromolaena tacotana*, flavanone, breast cancer, MDA-MB-231 and MCF-7 cell lines, apoptosis, XIAP, Bcl-2

## Abstract

*Chromolaena tacotana* is a source of flavonoids with antiproliferative properties in human breast cancer cells, the most common neoplasm diagnosed in patients worldwide. Until now, the mechanisms of cell death related to the antiproliferative activity of its flavonoids have not been elucidated. In this study, a novel flavanone (3′,4′-dihydroxy-5,7-dimethoxy-flavanone) was isolated from the plant leaves and identified by nuclear magnetic resonance (NMR) and mass spectrometry (MS). This molecule selectively inhibited cell proliferation of triple-negative human breast cancer cell lines MDA-MB-231 and MCF-7 whit IC_50_ values of 25.3 μg/mL and 20.8 μg/mL, respectively, determined by MTT assays with a selectivity index greater than 3. Early and late pro-apoptotic characteristics were observed by annexin-V/7-AAD detection, accompanied by a high percentage of the Bcl-2 anti-apoptotic protein inactivated and the activation of effector Caspase-3 and/or 7 in breast cancer cells. It was verified the decreasing of XIAP more than Bcl-2 anti-apoptotic proteins expression, as well as the XIAP/Caspase-7 and Bcl-2/Bax complexes dissociation after flavanone treatment. Docking and molecular modeling analysis between the flavanone and the antiapoptotic protein XIAP suggests that the natural compound inhibits XIAP by binding to the BIR3 domain of XIAP. In this case, we demonstrate that the new flavanone isolated from leaves of *Chomolaena tacotana* has a promising and selective anti-breast cancer potential that includes the induction of intrinsic apoptosis by downregulation of the anti-apoptotic proteins XIAP and Bcl-2. New studies should deepen these findings to demonstrate its potential as an anticancer agent.

## 1. Introduction

Flavonoids are a large group of natural polyphenolic compounds widely known for their beneficial effects on human health, due to their antioxidant, anti-inflammatory, anti-mutagenic and anti-cancer properties [1]. Flavonoids can be classified into different subgroups based on the degrees of unsaturation and oxidation of the C ring. Among these subgroups, flavanones, also called di-hydroxyflavones, have a saturated and oxidized C ring [2].

Flavanones are associated with several health benefits due to their free radical scavenging properties [1]. However, the research on their anticancer properties against breast cancer (BC) needs to be increased. An in-silico study disclosed the capacity of the flavonoids naringenin and hesperetin to inhibit the HER2-TK, in addition to their in vitro antitumor activity against HER2-positive tumors [3]. Pinocembrin, another promising natural flavanone for BC, has been isolated from several plants and induces cell cycle arrest of MCF-7, SKBR3, and MDA-MB-231 cells in G2/M phase by downregulation of the pro-survival proteins as cyclinB1, Cdc2 and Bcl-2 [4].

Bcl-2 family entails anti-apoptotic but also pro-apoptotic proteins, that interact to regulate mitochondria-based apoptosis [5]. The pro-survival Bcl-2 protein is frequently overexpressed in many tumors including BC and supports cancer cell growth and proliferation, tumorigenesis and chemoresistance. The active form of Bcl-2 is associated with phosphorylation at Serine 70 (S70pBcl-2) to prevent apoptosis by oxidative stress-induced DNA damage [6]. Active Bcl-2 protein interacts with the pro-apoptotic proteins Bak and Bax, preventing their heterodimerization and the formation of pores in the outer mitochondrial membrane, regulating the liberation of cytochrome-c into the cytoplasm [5,7]. On the other hand, the inhibitors of apoptosis proteins family (IAPs) confer BC survival and chemoresistance [8]. The X-linked inhibitor of apoptosis protein (XIAP), the most potent member of the IAP family, can neutralize the apoptotic function of Caspase-9 through the BIR3 domain, and effector Caspases 3 and 7 through the BIR2 domain [9] interfering with both the intrinsic and extrinsic pathways apoptosis.

The search for inhibitors or natural agents to interrupt protein-protein interactions between anti-apoptotic and pro-apoptotic proteins becomes an opportunity to explore new strategies for cancer treatment. In this respect, only a few studies relate to the regulation of anti-apoptotic proteins after the treatment of breast cancer cells with flavonoids. Quercetin is one of the most studied flavonoids and it is known that quercetin downregulates XIAP and sensitizes the cisplatin-induced apoptosis in oral squamous cell carcinoma (OSCC) [10].

*Chromolaena* species are distributed in temperate regions of Africa, Latin America, southern Asia, and Australia (www.theplantlist.org) (accessed on 9 November 2022). In ethnomedicine, they have been used in diseases such as malaria, nasal congestion, inflammation, eye disorders, asthma, cough, flu, headache, and cold, and more recently in cancer [11,12]. More than 190 compounds have been isolated and identified from 27 species of the genus, including flavonoids, alkaloids, triterpenoids, diterpenoids, sesquiterpenoids, steroids, fatty acids, and coumarins among others [13].

Around 12 flavanones of the *Chromolaena* genus have been isolated until now. From *C. odorata*: 5,3′-dihydroxy-7,6′-dimethoxyflavanone (Odoratenin), 5,3′-dihydroxy-7,4′-dimethoxyflavanone (persicogenin), 5,7-dihydroxy-4′-methoxyflavanone (Isosakuranetin), 5,7-dihydroxy-6,4′-dimethoxyflavanone, 4′-hydroxy-5,6,7-trimethoxyflavanone, 5-hydroxy-7,4′-dimethoxylflavanone and 5,6,7,4′-tetramethoxyflavanone, from *C. leivensis*: 5,7-dihydroxyflavanone (Pinocembrin) and 3,5-dihydroxy-7-methoxyflavanone, and 5,4′-dihydroxy-6,7-dimethoxyflavanone, 5,3′,4′-trihydroxy-7-methoxyflavanone and 4′,5,7-trihydroxyflavanone (Naringenin) from *C. subscandens, C. farinose* and *C. connivens*, respectively [13].

The endemic Colombian plant *Chromolaena tacotana* (Klatt) R. M. King and H. Rob (*C. tacotana*) is considered a source of flavonoids with demonstrated free radicals scavenging activity [14]. Furthermore, cytotoxic activity against colon cancer cell lines HT29 and RKO, prostate PC3, and triple-negative breast MDA-MB-231 [15,16]. This study aimed to report the chemical characterization and explore the anti-breast cancer properties of the new compound identified as 3′,4′-dihydroxy-5,7-dimethoxy-flavanone (named here Tacotanina) isolated from the leaves of *C. tacotana*.

## 2. Results

### 2.1. Structural Identification of the Flavanone

The flavanone was isolated from dichloromethane leaves extract of *C. tacotana* as described in Section 4.1. The chemical structure of the flavanone was obtained from the analysis of the UV spectra with displacement reagents (AcONa, MeONa, and H3BO3), ^1^H and ^13^C Nuclear Magnetic Resonance experiments and High-Resolution Mass Spectrometry. The compound was obtained as a yellow solid in the form of needles, soluble in DMSO, melting point of 179 °C, Rf of substance 0.25 (silica gel, CHCl_3_: MeOH 9.5: 0.5) fluorescent yellow spot when observed at UV at λ = 366 nm and in NH3 vapors it is purple. For the elucidation of the molecular structure of this compound, the analysis of the UV nm, ^1^H and ^13^C NMR and two-dimensional NMR (COSY, HSQC, and HMBC) spectra (Table 1 and Table 2) were registered as follow: according to UV nm: in MeOH 284, 388; plus MeONa 284, 388, 419; plus AcONa 284, 388; plus, H_3_BO_3_ does not present changes, as with the other displacement reagents. For the acetylated compound, the ^1^H NMR presents a singlet at 2.25 ppm which integrates for 6H, two acetate groups, which confirm the two free -OH groups in the compound. In this case, ^13^C JMOD NMR data confirm that the compound is one flavanone with two methoxy groups, two free OH, and without OH at C-5. The ring B has two oxygenated carbons (Appendix A) and considering the connectivity of the different hydrogens with the corresponding carbons (Figure 1), here it is demonstrated that the compound corresponds to 3′,4′-dihydroxy-5,7-dimethoxyflavanone.

The exact calculated mass corresponded to a 316.09019 +/− 1.42 ppm obtained from the ESI negative ion mode, [M − H]^−^: 315.08745 and positive ion mode [M + H]^+^: 317.10020, which suggested the molecular formula C_17_H_16_O_6_ (calc. 316.09469) (Appendix A).

The mass fragmentation analysis showed the loss of 110 Da by the cleavage of the 3′,4′-dihydroxy phenyl group. The ion formed is relatively easier to form and can be directly observed in [M + H]^+^ 207. On the other hand, the ESI negative ion mode in low collision energy (CE-4 eV) produced a prominent [M − H]^−^: ion at *m*/*z* 315.0875 and in a high CE (70 eV), the flavanone produced fragment ions at *m*/*z* 301 and 287 probably due to the loss of two methyl groups by the rearrangement of hydrogen to form two hydroxyl groups. The Retro Diels-Alder reaction (RDA) occurred and produced *m*/*z* 135 ions. By its MS^2^ frag-mentation, a methylene group was lost forming the *m*/*z* 107 ion, followed by the loss of water to give the *m*/*z* 89 ion (Appendix A).

### 2.2. Cytotoxic Activity and Selectivity Index on BC Cells

The flavanone Tacotanina and positive controls quercetin (Q) and paclitaxel (PTX) were evaluated for their cytotoxic potential against human BC cell lines MDA-MB-231 and MCF7, and normal breast MCF-12F and lung MRC5 cells. The results of the cell viability assay showed that Tacotanina inhibited 50% of the cell viability of breast cancer cells at concentrations between 56.9 and 72.7 μM, while normal cells at higher concentrations between 234 and 246.6 μM. Interestingly, Tacotanina had a better potential against MDA-MB-231 triple negative BC cells (Figure 2C) than MCF-7 cells (Figure 2D), which corresponds to a significant difference *p* < 0.05 (Figure 2E). Additionally, a significant difference *p* < 0.05 was found between the results obtained in the treatments with Q and Tacotanina in the two BC lines evaluated, with the cytotoxic potential of flavanone being better (Figure 2E). Regarding the selectivity index calculated with the cytotoxicity values of the compounds on normal cells (Figure 2A,B) and cancer cells (Figure 2C,D), Tacotanina showed a higher selectivity for BC cells than the positive controls, in fact, PTX was the most aggressive compound for normal cells (Figure 2F).

### 2.3. Apoptosis Induction

In order to evaluate the Tacotanina apoptosis-induced flow cytometric method using PTX as a positive control was used. Initially the Annexin V binding to surface exposed phosphatidyl-serine (PS), and the DNA dye 7-AAD to discriminate between early and late apoptosis in BC and normal cells. The data confirm the high cytotoxicity in normal cells exposed to PTX, leading more than 50% of MCF-12F cells to cell death by apoptosis (Figure 3C). Regarding the effect of Tacotanina, the induction of apoptosis was evidenced in more than 60% of the TNBC cells (Figure 3A), about 50% of the MCF-7 cells (Figure 3B), and less than 20% in the MCF-12F cells (Figure 3C) at 24 h of treatment at IC_50_. Discriminating early and late apoptosis, the compound achieved a higher number of TNBC cells in late apoptosis than in MCF-7 cells, 20.15 and 9.2%, respectively, while the early apoptosis induced was similar in both BC cells about 42%.

In order to test whether the induction of apoptosis depended on Caspases 3 and 7, the activation of these enzymes was evaluated in both BC cells at 24 and 48 h (Figure 3D,E). The dot plots obtained from BC cells exposed for 24 h to treatment with Tacotanina show the activity of Caspases 3 and/or 7 in both cell lines, which increases over time in consonance with the results obtained at 48 h where the activation of late apoptosis was evident after detection of the Caspases in combination with 7-AAD dye in more than 60% of cells (Figure 3F).

### 2.4. Morphological Analysis of Nuclei and Microtubules of MDA-MB-231

Microtubules and nuclei of MDA-MB-231 cells were evaluated by epifluorescence microscopy after treatment with PTX and Tacotanina. It was observed that the positive control PTX, a recognized microtubule damage agent, caused the disruption of microtubule dynamics as part of the mechanism of inhibition of cell proliferation and subsequent apoptosis, characterized by the condensation of nuclei and the apparent formation of apoptotic bodies (Figure 4). Similarly, treatment of TNBC cells with Tacotanina, at the IC_50_ value for 24 h, results in impaired microtubule integrity (white rows, Figure 4) and nuclei condensation (red rows, Figure 4).

### 2.5. Anti- and Pro-Apoptotic Proteins Expression and Complexes Dissociation

Western blot showed that treatment of MDA-MB-231 cells with Tacotanina induced a significant decrease in the level of the antiapoptotic XIAP protein evident after 24 h of treatment, while the expression of Bcl-2 decreased slightly until 24 h, and then a more elevated level was observed concluding that there is not a large difference concerning to treated cells (Figure 5A) while the pro-apoptotic Bax protein expression increases at 48 h.

The antiapoptotic complex between Bcl-2 and Bax proteins was interrupted after treatment according to co-immunoprecipitation results (Figure 5B). Concerning to cleaved-casp-3, this active form increases in quantity after treatment, as occurred in cells treated with PTX (Figure 5B), and Western blot after coimmunoprecipitation showed a dissociation between Casp-3 and the antiapoptotic XIAP protein in a time-dependent manner.

### 2.6. In Silico Analysis with the Antiapoptotic Protein XIAP

In order to establish whether Tacotanina could bind to the XIAP-BIR3 domain in a similar way to previous natural compounds and act as a potential inhibitor of XIAP, we assessed the molecular docking between Tacotanina and the BIR3 domain of XIAP. After the docking experiments, the binding site of Tacotanina with the BIR3 domain of XIAP, fits in to the pockets P1-P4 (Figure 6A) previously reported in different complex with Smac and embelin [17]. In addition, Tacotanina showed interactions of hydrogen bonds with residues THR 308, LYS 311, GLU 314, and TRP323, which are involved similarly way in the interaction of the BIR3 domain of XIAP and Smac or Smac mimetics [18].

In order to establish the stability of the protein-ligand complex formed between Tacotanina and the BIR3 domain of XIAP, molecular dynamics simulations were performed with a production of 50 ns (Figure 6B,D). The results showed that the complex was stable during the production time of the molecular dynamics simulation with slight variations in the RMSD trajectory (Figure 6D). Indeed, showed that the binding mode obtained in the molecular docking analysis and molecular dynamics simulations are according with other results reported in different computational experiments for another small molecules antagonist of XIAP and suggest that Tacotanina can be able to form a highly stable complex with the BIR-3 domain of XIAP [19,20].

### 2.7. Bcl-2 Regulation

In order to assess the ability of Tacotanina to deregulate S70pBcl-2 in BC cells, a flow cytometry assay was performed using two antibodies, one of them to measure the phosphorylated state of Bcl-2 at Ser70 and the other to detect non-phosphorylated protein or inactive. In approximately 90% of the untreated cells, the S70pBcl-2 protein was detected, but after treatment with Tacotanina, the number of S70pBcl-2 cells decreased progressively until only 11 and 18% of the MDA-MB-231 cells and MCF-7, respectively, expressed the active protein. In BC cells exposed to PTX for 24 h, the results showed that S70pBcl-2 was decreased in a low percentage of cells, only up to 48 h in TNBC cells the protein was inactivated up to 82%, but in MCF-7 cells, the inactivation was not reached to 50% (Figure 7).

## 3. Discussion

*Chromolaena* is characterized by the presence of the flavonoid family of compounds, of which 79 have been isolated and chemically identified [13]. This phytochemical investigation in *C. tacotana* led to the isolation of a new flavanone no reported before in any plant species. The compound has an experimental molecular weight of 316.09019 (C_17_H_16_O_6_, calc. 316.09469) and according to the ^13^C APT NMR spectrum, 17 signals were identified, six correspond to methane-type carbons (CH), and two more to methyl groups (CH_3_) corresponding to methoxyl groups in a negative phase. In the positive phase, the methylene group (CH_2_) is differentiated at δ = 45.20 ppm. The remaining eight signals correspond to common quaternary carbons from the aromatic ring and sp^2^ carbons oxygenated at a displacement of δ = 162.18, 164.83, and 165.79 ppm, as well as a signal at δ = 188.59 ppm of the assignable carbonyl functional group for the C-4. Structurally, the location of the hydroxyl groups on the B-ring of flavanone was evidenced by the loss of the 5,3ʹ-dihydroxy phenyl group (*m*/*z* 110), identified in the positive mode fragmentation of the *m*/*z* 317 molecular ion, giving rise to the *m*/*z* 207 ion. This result is also evidenced by the formation of the *m*/*z* 135 ion, which by the loss of methylene group after RDA cleavage, and loss of water, giving rise to the *m*/*z* 89 ion, in the negative ionization mode. A similar fragmentation was identified in the study of the flavanone Liquiritigenin, which has a molecular ion [M + H]^+^ 255 with fragment ions at *m*/*z* 135 and 119, inferring a retro Diels-Alder cleavage. The *m*/*z* 119 ion then loses a methylene group obtaining an *m*/*z* 91 ion [21].

Evaluating the cytotoxic activity and selectivity of Tacotanina isolated from *C. tacotana* on BC cells, we found that it is capable of inhibiting cell viability with high selectivity for cancer cells at concentrations between 56.9 and 72.7 μM, which is higher compared to the flavanone previously isolated from *C. tacotana* the 5,7-dihydroxy flavanone that inhibited cell viability and proliferation of colon adenocarcinoma (HT29), prostate cancer (PC-3), adenocarcinoma human alveolar basal epithelial (A549), breast adenocarcinoma (MDA-MB-231), and cervical squamous carcinoma (SiHa) cell lines [15], and another flavanones from *C. leivensis* the 3,5-dihydroxy-7-methoxyflavanone that showed mild cytotoxicity against the colon cell line (CaCo-2) (EC_50_ = 120.80 uM) [22] and had activity on the mitochondrial membrane at varying concentrations (25, 50 and 100 μg/mL) [23]. Cytotoxicity of Tacotanina is even better than the activity found for other flavanones considered to be the main or most studied so far, 3′,5,7-trihydroxy-4′-methoxyflavanone (hesperetin), which previously showed an IC_50_ value of 129 and 87 μM for MDA-MB-231 and MCF-7, respectively [24], and 4′,5,7-trihydroxyflavanone (naringenin) that inhibits the viability of MDA-MB-231 cells above 146.9 μM [25]. The suitable number and positions of hydroxyl groups, particularly the ortho-hydroxyl system on B-ring have been associated with potent antiproliferative activity [26], which is a feature of Tacotanin but not of hesperetin, the naringenin or the other cytotoxic flavanones of the *Chromolaena* genus.

These findings prompt the study of the mechanism of cell death of Tacotanina. Apoptosis induction was detected by annexin V in the early event that involves the plasma membrane bilayer changes and exposure of phosphatidylserine (PS) on the outer surface. The early apoptosis was detected in both MDA-MB-231 and MCF-7 cells at IC_50_, but not in normal MCF-12F cells, confirming the selectivity shown by the viability cell assay [27]. In addition, the DNA intercalation dye 7-AAD (7-amino-actinomycin D), allowed us to differentiate between early, late, and possibly necrotic apoptotic cells. Our findings after 24 h of treatment with Tacotanina demonstrate the induction of late apoptosis in about 20% of MDA-MB-231, twice as much as in MCF7 cells, confirmed by morphological alterations such as cytoskeleton disorganization due to loss of microtubule integrity, an apoptotic event induced by microtubule-targeting agents (MTAs) such as PTX, which can bind to tubulin to alter microtubule dynamics by polymer stabilization, affecting cell division and inducing cell death of cancer cells [28]. Likewise, the fragmentation and condensation of nuclei and the subsequent formation of apoptotic bodies were observed (Figure 4).

As the Caspases 3 and 7 are considered to be enzymes that play a key role at the end of the intrinsic and extrinsic pathways of apoptosis induction [29], we evaluated its activation by flow cytometry, observing that of the 42% of cells in early apoptosis (Figure 3A,B), the majority, around of 36%, have active Caspases 3 and 7 in MDA-MB-231 cells (Figure 3D), while a smaller number of MCF-7 cells, near to 29%, were detected with active Caspases (Figure 3E), which surely correspond to Caspase-7 since it was previously reported that Caspase-3 is not expressed in MCF-7 cells and that Caspase-7 could compensate for some of its functions during apoptosis [30]. From these results, it could be inferred from the data obtained after exposure of MCF-7 cells to Tacotanina, it occurs independently of Caspase-3activity.

Inhibition of apoptosis by the X-linked inhibitor of apoptosis protein (XIAP) is one of the leading causes of tumor cells growth and the high expression of XIAP is frequently one culprit in different types of cancer and plays an important role in developing chemoresistance. The principal role of XIAP as an antiapoptotic protein is to inhibit active Caspases 9, 7, and 3, through its BIR2 and BIR3 domains. The inhibition of the BIR3 domain of XIAP by small molecules has been considered an emerging therapeutic target in the discovery of new anti-cancer therapeutics based on promoting apoptosis in cancer cells [31,32]. Western blot analysis revealed that MDA-MB-231 breast cancer cells treated with Tacotanina, decreased the levels of XIAP at 24 h while the levels of cleaved-casp-3 were enhanced, such as occurred in cells treated with PTX (Figure 5A). Interestingly, the co-immunoprecipitation analysis showed a dissociation in a time-dependent manner of the complex formed between the Caspase-3and the antiapoptotic XIAP protein (Figure 5B).

A molecular modeling approach with the BIR3 domain of XIAP has demonstrated that some natural molecules such as embeline, eriopodol A, gibbilimbol B, and erioquinol, can bind to the BIR3 domain of XIAP in the same binding site of Smac (the endogenous antagonist of XIAP) and promote apoptosis in different types of human cancer cells [32,33]. Our predicted results obtained by molecular modeling suggest the binding mode of Tacotanina to the BIR3 domain of XIAP (Figure 6A,B), and are consistent with the downregulation of XIAP and subsequent enhanced levels of cleaved-Caspase-3 observed in breast cancer cells treated with Tacotanina to promote apoptosis (Figure 5).

However, activation of Caspase-3 requires cleavage of pro-Caspase-9 in a process involving pro-apoptotic proteins of the Bcl-2 family of the intrinsic pathway. In this pathway, Bax heterodimerizes to form oligomers, creating pores in the outer mitochondrial membrane (OMM) allowing cytochrome-c to release into the cytoplasm which mediates the allosteric activation of apoptosis-protease activating factor 1, required for the proteolytic cleavage of Caspase-9 to active the Caspase-3 [5]. In this respect, we have found an increase in Bax protein levels (Figure 5A) as well as the anti-apoptotic complex dissociation after treatment with the flavonoid, similar to occur with the PTX control (Figure 5B), indicating that Tacotanina induces intrinsic apoptosis as a response to treatment in TNBC cells.

Expression levels of the antiapoptotic protein Bcl-2 did not change significantly during treatment with the flavonoid (Figure 5A), but it has been mentioned that Bcl-2 regulation can occur independently of its expression levels and can depend on the phosphorylation of the protein at the site within the flexible loop domain (FLD) required for the antiapoptotic function of Bcl-2 [34]. The treatment with Tacotanina results in the decrease of a significant percentage of the post-translational modification of Bcl-2 in serine 70 (Figure 7), maybe by activation of mitochondrial enzyme PP2A responsible for dephosphorylation. In contrast, analyzing the apoptosis induced by PTX, we found that this effect occurs independently of Bcl-2 protein dephosphorylation. Reports indicated that antimitotic drugs such as paclitaxel can inhibit Bcl-2 by phosphorylation at multiple sites, which in addition to serine 70 include threonine 69 and serine 87, indicating that phosphorylation does not occur exclusively at serine 70 [34].

## 4. Materials and Methods

### 4.1. Extraction and Isolation

The leaves of *C. tacotana* were collected in La Vega, in the Department of Cundinamarca (Colombia). A voucher sample was deposited at the Herbarium of Pontificia Universidad Javeriana, Bogotá, Colombia, and determined as *Chromolaena tacotana* with a voucher number HPUJ30170.

All reagents were analytical grade and obtained from Merck (Rahway, NJ, USA). 403 g of dried and ground leaves were subjected to maceration with ethanol 96% to obtain 94 g of extract. After the extract was submitted to a Soxhlet extraction with dichloromethane (DC) CH_2_Cl_2_ to remove the content of fats and chlorophylls, next 35.8 g from that total extract named (DC-EI) was flocculated with methanol (MeOH): water (1:1), and after, the aqueous portion was extracted with CH_2_Cl_2_ and concentrated in vacuum. This second dichloromethane extract was named (DC-EII) and it was used to obtain flavonoids. 30.7 g from DC-EII were separated by column chromatography with Silica gel (40–60 μm) and RP18 (20–40 μm), the flavonoids were isolated using a mixture of CHCl_3_: MeOH in a ratio of 9.8: 0.2 for the flavanone and crystallization was performed with n-Hexane to obtain the flavanone (11.9 mg). Identification (Table 1) was carried out using UV (nm) spectra taken on a Jenway 6405 UV-VIS spectrophotometer with displacement reagents (AcONa, MeONa, and H_3_BO_3_, Merck, Rahway, NJ, USA).

Mono and two-dimensional ^1^H NMR and ^13^C NMR spectra were recorded on an Avance Bruker 300 spectrophotometer at 300 MHz for 1H and 75 MHz for ^13^C, using the solvent peaks as internal references, the spectra were recorded in Acetone-d6 (Merck, Rahway, NJ, USA). High-resolution mass data were collected using ultra-high-pressure liquid chromatography (UPLC) coupled to a quadrupole time-of-flight (Q-TOF) mass spectrometer detector (QTOF), Shimadzu (LCMS-9030).

### 4.2. Cell Lines and Culture

The human breast cancer cell lines MDA-MB-231 (HTB-26) and MCF7 (HTB-22), were grown in RPMI 1640 (Lonza, SC, USA) and Eagle’s Minimum Essential Medium–EMEM (Lonza, SC, USA), respectively, both supplemented with 10% heat-inactivated fetal bovine serum (Biowest, Nuaillé, France) and 1.0% of penicillin/streptomycin (Lonza, SC, USA), the normal MRC5 (CCL-171) fibroblast cells were also grown in supplemented EMEM, and the normal epithelial MCF-12F (CRL-10783) of mammary gland cells were grown in DMEM/F-12 (Sigma-Aldrich, St Louis, MO, USA) supplemented with 7.0% fetal horse serum (Sigma-Aldrich), 10 μg/mL human insulin (Sigma-Aldrich, St Louis, MO, USA), 20 ng/mL epidermal growth factor (Sigma-Aldrich, St Louis, MO, USA), 500 ng/mL hydrocortisone (Sigma-Aldrich, St Louis, MO, USA), 100 mg/mL cholera toxin (Sigma-Aldrich, St Louis, MO, USA), and 1.0% of penicillin/streptomycin (Lonza, SC, USA). Cells were incubated at 37 °C and 5.0% CO_2_ in a humidified atmosphere.

### 4.3. Cytotoxic Activity and Selectivity of Flavanone

Cell survival in response to treatments was determined by 3-(4,5-methyl-thiazol-2-yl)-2,5-diphenyl-tetrazolium bromide (MTT) assay, cells were seeded at a density of 7000 cells per well onto 96-well plates and incubated under a humidified environment at 37 °C and 5.0% CO_2_ for 24 h. Cells were treated with the flavanone dissolved in dimethyl sulfoxide (DMSO) (Sigma-Aldrich, St Louis, MO, USA) at concentrations between 160 and 1.5 μg/mL per triplicate. The maximum final concentration of DMSO was 0.5% per treatment. After 48 h of incubation, 100 µL of 500 μg/mL MTT was added (Sigma-Aldrich, St Louis, MO, USA) and then incubated for additional 4 h. Formazan crystals were solubilized with DMSO, and absorbance values were measured at 570 nm using a microplate reader (BioRad, Hercules, CA, USA). The inhibitory concentration required to decrease 50% of cell viability (IC_50_) concerning to negative control untreated cells, was estimated by non-linear regression using GraphPad Prism 8.0 (La Jolla, CA, USA). The selectivity index was calculated by the relation between IC_50_ normal cells/IC_50_ tumor cells [35].

### 4.4. Apoptosis Assay

Detection of live cells and cells in early or late apoptosis was performed using Muse^®^ Annexin V & Dead Cell Reagent Kit (Luminex Corporation, IL, USA). Approximately 40,000 cells were seeded per well in a 24-well plate, to be 60–70% confluent, and after 24 h were treated with Paclitaxel and 3′,4′-dihydroxy-5,7-dimethoxyflavanone at the corresponding IC_50_ for 24 h. Cells with 0.5% DMSO molecular grade (Thermo-Fisher Scientific, Waltham, MA, USA) were used as a negative control. The cells were collected and prepared by adding 100 µL of Annexin-V/7-AAD reagent to each tube containing 100 µL of cell suspension (500 cells per µL) and incubated for 20 min at room temperature in the dark. Subsequently, the samples were measured on the Guava^®^ Muse^®^ Cell Analyzer (Luminex Corporation, IL, USA) and the Muse^®^ software 1.8. The induction of apoptosis was confirmed by the detection of active Caspase 3/7 enzymes. Initially, 40,000 cells per well were seeded in a 24-well plate. After adherence, the treatment was performed with the flavonoid and the positive control Paclitaxel at the IC_50_ determined for each cell line and the negative control with 0.5% DMSO for 24 h and 48 h. After treatments, both, the medium and the harvested cells were centrifuged and resuspended with the 1X assay buffer of the Muse^®^ Caspase-3/7 Kit (Luminex Corporation, IL, USA), according to the manufacturer’s instructions. To the cell suspension, 5 µL of Muse Caspase-3/7 reagent working solution was added to each sample. Afterward, the samples were incubated without being completely covered for 30 min at 37 °C with 5.0% CO_2_. Later 150 μL of Muse Caspase 7-AAD reagent solution was added to each sample and incubated in the dark at room temperature for 5 min. Finally, the lives cells and the cells in apoptosis with active Caspase-3/7 proteins were detected on the Guava^®^ Muse^®^ Cell Analyzer (Luminex Corporation, IL, USA) and the Muse^®^ software 1.8.

### 4.5. Morphological Analysis in Nuclei and Microtubules by Epifluorescent Microscopy

In 24-well plates, the TNBC cells were fixed at 60–70% confluency, previously treated with Tacotanina and PTX to the IC_50_ of each compound determined by the MTT assay, and incubated at 37 °C with 5.0% CO_2_ for 24 h. Next, the staining processes will be carried out, using DAPI to label DNA and the anti-alpha tubulin antibody T9026 (Sigma-Aldrich, St Louis, MO, USA) as a microtubule marker and Alexa fluor 388 (FITC). The images were captured with MoticCamPro 282A and analyzed at 40X using Motic Image plus 2.0 software (Motic, Kowloon, Hong Kong) [36].

### 4.6. Western Blot Analysis of Pro- and Anti-Apoptotic Proteins

After the treatments with Tacotanina and PTX for 16, 24, and 48 h, the MDA-MB-231 cells were collected. Lysis buffer (20 mM Tris HCl pH 8.0, 137 mM NaCl, 10% glycerol, 1.0% NP-40, and 10 mM EDTA) was used to obtain a total protein extract of each sample. To measure the concentration of the total protein the bicinchoninic acid (BCA) assay (Pierce) was used. In this case, 30 μg of protein was loaded in 11% SDS-PAGE. Proteins were blotted using polyvinylidene fluoride (PVDF) membranes (Thermo-Fisher Scientific, Waltham, MA, USA), subsequently blocked for 1 h with 5.0% (*w*/*v*) BSA/TTBS and incubated overnight at 4 °C with the primary antibodies: anti-Bcl-2 (N1N2) (GeneTex, Irvine, California, USA), anti-cleaved-casp-3 (Thermo Fisher Scientific, Waltham, MA, USA), anti-XIAP (D2Z8W), anti-Bax (D2E11) anti-COX IV (3E11) (Cell signaling) the latter was used as the loading control. The next day, the membranes were incubated for 1 h at room temperature, with the corresponding secondary antibody, either anti-mouse IgG or anti-rabbit IgG (Merck Group, DE, Darmstadt, Germany) [35]. Bands detection was performed by using the chromogenic HRP system 1-Step™ TMB-Blotting Substrate Solution (Thermo-Fisher Scientific, Waltham, MA, USA) yields a blue-colored precipitate. ImageJ software was used to normalize and semi-quantify bands’ density.

### 4.7. Co-Immunoprecipitation of Anti and Pro-Apoptotic Proteins Complexes

In this case, 200 μg of each total protein sample from untreated and treated proteins were mixed with 2 μg of anti-Bcl-2 antibody (Genetex, Irvine, CA, USA) or anti-XIAP antibody (Cell signaling) in a dilution buffer (50 mM Tris HCl pH 7.4, 0.5 mM EDTA, 1.0% NP40, 50 nM NaF, 0.5 mM PMSF, and protease inhibitors cocktail (1:100) (Pierce, WA, USA). Samples were incubated at 4 °C for 4 h and then incubated with 40 ul protein G agarose (Thermo-Fisher Scientific, Waltham, MA, USA) at 4 °C overnight, and then washed three times with the buffer (50 mM Tris HCl pH 7.4, 250 mM NaCl, 0.5 mM EDTA, 1.0% NP40, 50 nM NaF, 0.5 mM PMSF, and protease inhibitors cocktail (1:100) (Pierce, WA, USA)). Here, 35 µL of 1× loading buffer was added and the samples were boiled for 5 min at 90 °C [35]. Detection of the pro-apoptotic Bax protein bound to Bcl-2 protein, or Caspase-3linked to XIAP protein, was performed using the protocols described above for SDS-PAGE and Western blot. However, XIAP/Casp-3 was detected by using Caspase-3Antibody (31A893) (NOVUS Biologicals, CO, USA) to detect the full-length of Caspase-3.

### 4.8. Phospho-Bcl-2 Protein Detection

The regulation of Bcl-2 protein by phosphorylation at serine 70 (S70pBcl-2) related to the induction of the intrinsic pathway of apoptosis [6], was analyzed by flow cytometry. In this case, 200,000 cells were seeded in a 12-well plate and after incubation overnight at 37 °C and 5.0%, CO_2_ adherent cells were exposed at IC_50_ of Tacotanina and PTX for 24 and 48 h. To measure the Bcl-2 protein we follow the instructions suggested in the Muse™ Bcl-2 Activation Dual Detection Kit which includes two directly conjugated antibodies, anti-phospho-Bcl-2 (Ser70)-Alexa Fluor^®^555 and an anti-Bcl-2-PECy5 to detect inactive Bcl-2 expression (Luminex Corporation, Austin, IL, USA). Treated and untreated cells were dissociated and collected for fixation, permeabilization, staining, data acquisition, and analysis on the Muse™ Cell Analyzer software 1.8. Three replicates were analyzed under the same conditions and results were presented as the mean of them.

### 4.9. Geometry Optimization

The chemical structure of 3′,4′-dihydroxy-5,7-dimethoxyflavanone, was drawn using ChemDraw [37]. The file (.sd) format was converted to the corresponding three-dimensional (3D) structure and saved as .pdb format using Open Babel [38]. Subsequently, the 3D structure was optimized using molecular mechanics with the MMFF94 force field from the software Avogadro [39].

### 4.10. Preparation of the XIAP BIR3 Domain

The 3D crystal structure of the BIR3 domain was obtained from PDB 5C83 and reported in the Protein Data Bank (Bank 2022). The ligand, water molecules, and metals were removed using PyMOL software 2.5 (DeLano Scientific LLC, CA, USA) (“PyMOL|pymol.org” 2022). Polar hydrogens were then added, nonpolar hydrogens were fused, and Kollman charges were added using MGLTools [40]. The prepared protein was saved in PDBQT format.

### 4.11. Molecular Docking

The AutoDock4 program (version 4.2.6) [41] was used to perform molecular docking [33] AutoGrid [41] was used through the MGLTools [42] interface [40] for the preparation of energy maps. The grid size was set to 50, 50, and 50 Å for x, y, and z, respectively. The Lamarckian Genetic Algorithm (LGA) was chosen to search for the best domain-flavonoid interaction conformations, with 2.5 million evaluations. The docking processes were performed with the default parameters of AutoDock 4. The best conformation obtained was used to perform molecular dynamics analysis.

### 4.12. Molecular Dynamics

Molecular dynamics simulations were performed using the academic version of Schrödinger’s simulation package Desmond V6.2 (Desmond Molecular Dynamics System; D. E. Shaw Research, New York, NY, USA, 2016) [43]. The domain-flavonoid structure was optimized using the Protein Preparation Wizard tool and the system was solvated in an orthorhombic box using the simple point charge model (SPC water model), as well as the predefined model for an electrically neutral system (physiological concentrations of monovalent ions, 0.15 M NaCl). Conditions were applied to the system in the NPT (constant number of particles, constant pressure, and temperature) set, the temperature was set at 300 K, the pressure was maintained at 1.01325 bar and the pH was maintained at 7.0 throughout the process. simulation, the force field OPLS_2005 was used. The production time in the simulation was 50 ns.

### 4.13. Statistical Analysis for Biological Test

The data were expressed as the mean of three biological replicates (n = 3) ± SD (standard deviation) and the difference between control and treatments were determined using a two-way ANOVA test and Tukey multiple comparison test. The statistical analysis test was performed using GraphPad Prism 8.0 (La Jolla, CA, USA). *p*-values were calculated to infer statistical significance (** *p* < 0.05 or *** *p* < 0.001).

## 5. Conclusions

This study reports a new flavanone identified as 3′,4′-dihydroxy-5,7-dimethoxy-flavanone isolated from *C. tacotana*, here named Tacotanina, which is highly cytotoxic and selectively on breast cancer cell lines. It can induce morphological changes related to apoptosis as a response to treatment, inactivation of anti-apoptotic Bcl-2 protein, and inhibition of XIAP through binding directly to its BIR3 domain. The treatment decreased Bcl-2: Bax and XIAP: Caspase-3 complexes ratio, increasing the cleaved-casp-3 protein levels in TNBC cells to promote the intrinsic apoptosis pathway. These novel findings suggest a promising natural agent against even the most resistant breast cancer cells, that should continue to be investigated in several cancer models.

## Figures and Tables

**Figure 1 molecules-28-00058-f001:**
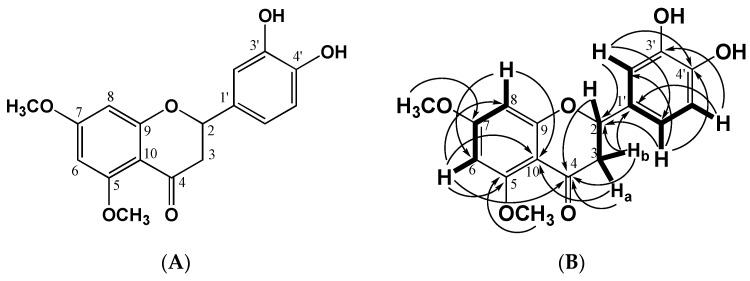
The structure of 3′,4′-dihydroxy-5,7-dimethoxyflavanone (Tacotanina). (**A**) Molecular structure. (**B**) Key COSY (bold) and HMBC (H → C) correlations for the compound.

**Figure 2 molecules-28-00058-f002:**
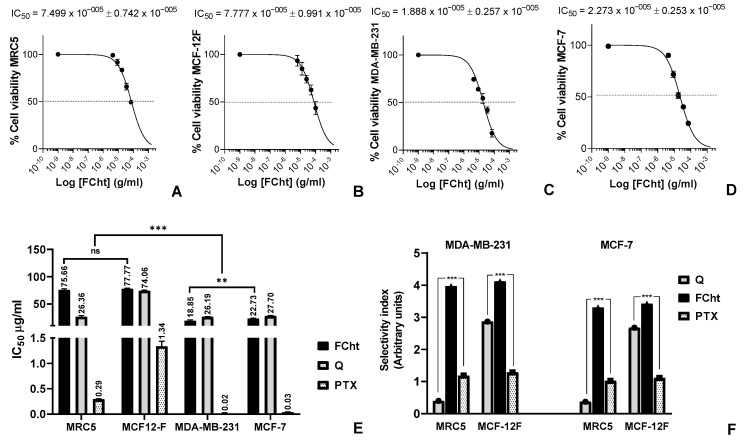
Cytotoxic activity and selectivity of Tacotanina on BC cells. (**A**) MRC5, (**B**) MCF12-F, (**C**) MDA-MB-231, (**D**) MCF-7, (**E**) half-maximal concentration (IC_50_), and (**F**) selectivity index on BC cells. Three independent experiments, each performed in triplicate were carried out and data were analyzed in GraphPad Prism 8.0, (La Jolla, CA, USA). *p*-values indicate statistical significance (** *p* < 0.05 or *** *p* < 0.001).

**Figure 3 molecules-28-00058-f003:**
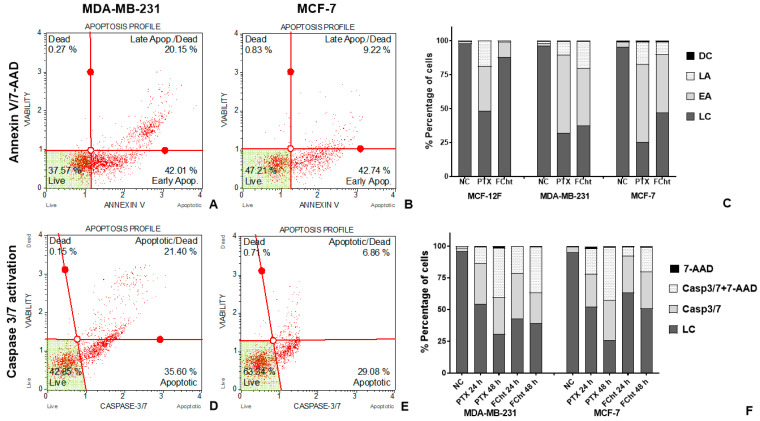
Apoptosis induction on BC cells. The dots (in red) represent a single cell analyzed, and dots in the green area correspond to live cells. (**A**) Annexin V/7-AAD detection on MDA-MB-231 cells exposed to Tacotanina, and (**B**) On MCF-7 cells. (**C**) Percentage of live cells (LC), cells in early apoptosis (EA), late apoptosis (LA), and non-apoptotic dead cells or nuclear debris (DC) after treatment with paclitaxel (PTX) or flavanone (Tacotanina) on MCF-12F, MDA-MB-231 and MCF-7 cells; NC corresponds to non-treated cells (**D**) Dot plot of Caspases 3/7 activation upon treatment with Tacotanina on MDA-MB-231 cells and (**E**) on MCF-7 cells. (**F**) Percentage of live cells (LC), early apoptotic cells with Casp3/7 activated, apoptotic cells with both Casp3/7 and 7-AAD detection, and non-apoptotic dead cells stained with 7-AAD after treatment with PTX or Tacotanina on MDA-MB-231 and MCF-7 cells; NC correspond to non-treated cells.

**Figure 4 molecules-28-00058-f004:**
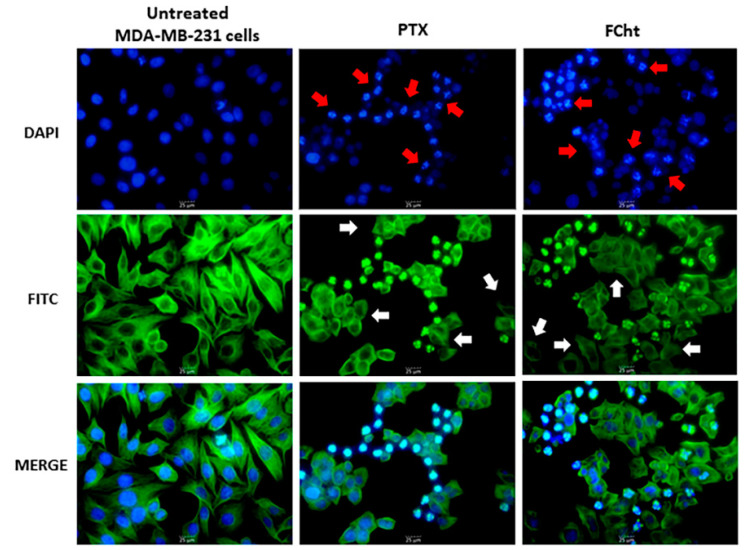
Microtubular and nuclear changes in MDA-MB-231 cells induced by Tacotanina treatment. Immunofluorescence microscopy images showing nuclear (in blue) and microtubular (in green) effects in TNBC cells. Cells treated with the positive control PTX showed microtubule damage and an apparent presence of apoptotic bodies. Cells treated with Tacotanina at 24 h were visibly affected in microtubule arrangement (white arrow) and nuclear condensation was observed (red arrow), resulting in the formation of apoptotic bodies.

**Figure 5 molecules-28-00058-f005:**
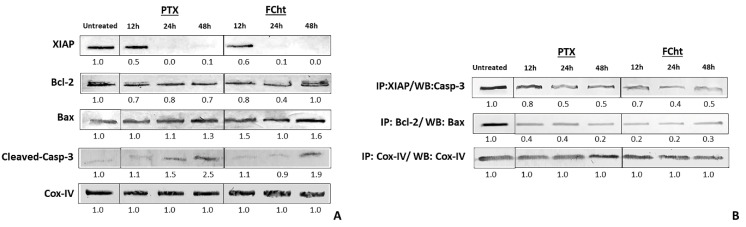
Western blot analysis of antiapoptotic XIAP and Bcl-2 and pro-apoptotic BAX and full-length-Caspase-3proteins and complexes on MDA-MB-231 cells. (**A**) Protein expression levels after treatment with Tacotanina and the positive control PTX for 12, 24, and 48 h. (**B**) Complexes’ levels resulting after treatment with Tacotanina and the positive control PTX for 12, 24, and 48 h. Co-IP the proteins immobilization on Protein G agarose was carried out with anti-XIAP or anti-Bcl-2 and Western blot with anti-Caspase-3 (31A893) and anti-Bax (D2E11), respectively. COX-IV mitochondrial protein was used as the loading control.

**Figure 6 molecules-28-00058-f006:**
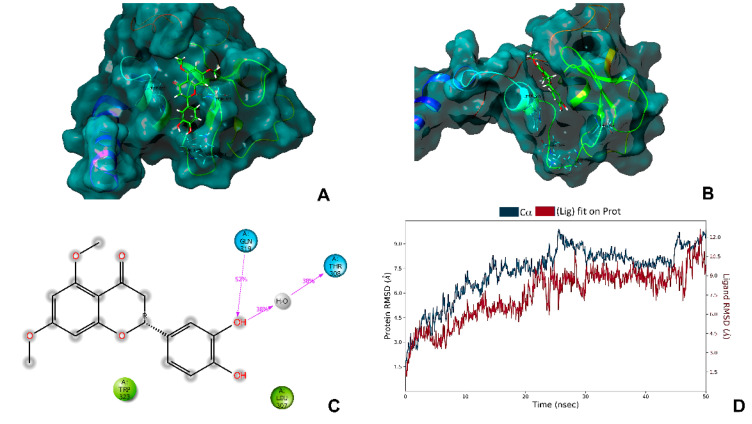
(**A**) Representation of the molecular docking and (**B**) the molecular dynamics complex after 50 ns of Tacotanina with the BIR3 domain of XIAP (pdb5c83). (**C**) Interaction Diagram 2D and (**D**) protein-ligand root mean square deviation trajectory of the atomic positions for ligands (red, Lig fit Prot) and the receptor (blu, C⍺ positions) of Tacotanina and the BIR-3 domain of XIAP, for the dynamic’s trajectory of 50 ns.

**Figure 7 molecules-28-00058-f007:**
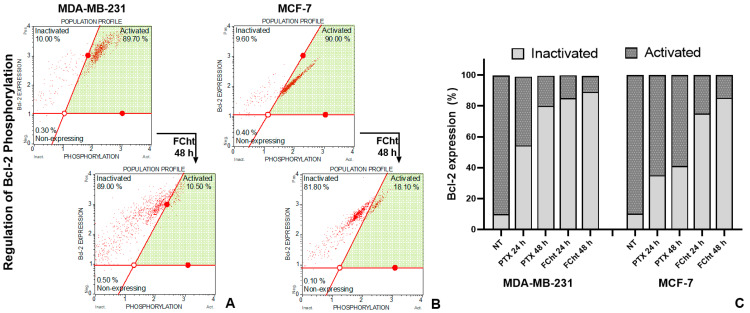
Phosphorylated Bcl-2 at Ser70 detection by flow cytometry on BC cells. The dots (in red) represent a single cell analyzed, and the green area correspond to region of active cells. (**A**) Dot plots comparing untreated MDA-MB-231 cells and cells exposed to Tacotanina at IC_50_ for 48 h. (**B**) Dot plots comparing untreated MCF-7 cells and cells exposed to Tacotanina at IC_50_ for 48 h. (**C**) Percentage of active and inactive Bcl-2 protein expression in BC cells treated with Tacotanina and the positive control PTX for 24, and 48 h.

**Table 1 molecules-28-00058-t001:** ^1^H NMR and ^13^C NMR (300 MHz, Acetone-d_6_) spectral data for the flavonoid Tacotanina.

Position	Compound Tacotanina	
	δH (J in Hz)	δC (ppm)
2	5.35 (1H, dd, J = 12.7; 3.31)	79.8
3a	2.94 (1H, dd, J = 16.3; 12.7)	46.2
3b	2.59 (1H, dd, J = 16.3; 3.0)	
4	-	188.3
5	-	163.2
5-OMe	3.85 (3H, s)	56.0
6	6.15 (1H, d, J = 2.3)	94.3
7	-	166.6
7-OMe	3.81 (3H, s)	56.1
8	6.17 (1H, d, J = 2.3)	93.4
9	-	165.2
10	-	106.6
1′	-	132.0
2′	7.02 (1H, brs)	114.6
3′	-	145.9
4′	-	146.2
5′	6.86 (1H, brs)	115.9
6′	6.86 (1H, d, J = 1.2)	119.1

**Table 2 molecules-28-00058-t002:** UV nm spectral data for the Flavanone Tacotanina.

Compound	UV nm	Supplementary Data
3′,4′-dihydroxy-5,7-dimethoxyflavonone (Tacotanina)	In MeOH 284, 388; plus MeONa 284, 388, 419; plus AcONa 284, 388; plus, H_3_BO_3_ does not present changes, as with the other displacement reagents.	For ^1^H NMR, ^13^C NMR, COSY, HMBQC, HMBC, and HRESIMS data, see Appendix A

## Data Availability

The data presented in this study supporting the results are available in the main text and Appendix A. Additional data are available upon reasonable request from the corresponding author.

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
