# Peer review of "A New Flavanone from *Chromolaena tacotana* (Klatt) R. M. King and H. Rob, Promotes Apoptosis in Human Breast Cancer Cells by Downregulating Antiapoptotic Proteins"

_molecules, 2022, doi:10.3390/molecules28010058_

Round 1

Reviewer 1 Report

The manuscript by Mendez-Callejas G et al. nicely describes the presence of a potential anticancer drug in the flavanones extracts obtained from the chromolaena taconata. The manuscript has been well designed and the experimental data allow authors to obtein solid conclusions. Further the use of different breast cancer lines strengh the study. Further the authors not only have demonstrated the negative effect in breast cancer cell proliferation, but they have deeply investigated the intracellular molecular pathways that may be affected by the tacotanina. Considering the partial specificity (having bigger effect in TNC cells than control of MCF7) and that they have defined the molecular target of this drug, I would endorse the publication of the manuscritp in its present form, although I would suggest that authors investigate the possible effects of tacotanina in different types of TNC cells. The latter may contribute to a better understanding of the relevance and possible translationary potential of the drug that they have just discovered. But I may undertand that author would address this suggestion in future studies.

Author Response

Dear reviewer, thank you very much for the positive comments and for the suggestions regarding the study, which indeed we must continue addressing to give this molecule a greater scope as an anticancer agent, this includes, among others, in vitro, in silico, and in vivo trials. We include this recommendation in the conclusion section.

Best Regards

Reviewer 2 Report

It is interesting to read the manuscript A new Flavanone Isolated from Chromolaena tacotana (Klatt) R. M. King and H. Rob, Promotes Apoptosis in Human Breast Cancer Cells by Downregulating Antiapoptotic Proteins”. I appreciate the authors' efforts in isolating a novel flavanone and studied about its anticancer activity against human breast cancer cell lines. This will be more beneficial to scholars working drug discovery and development in breast cancer therapy using natural products. However, if the following changes are made and they are incorporated into the manuscript, it could be considered for publication in “Molecules” Journal.

 1.    Avoid “Our findings/study/predicted” throughout the manuscript. Instead use “The present/current study”…I advise authors to modify the title as “A new Flavanone from Chromolaena tacotana (Klatt) R. M. King and H. Rob, Promotes Apoptosis in Human Breast Cancer Cells by Downregulating Antiapoptotic Proteins”.

2.    Revise the conclusion in the abstract. In the conclusion, I suggest that authors to offer a critical justification on their findings and observations. Further to be extended with one line about future direction.

3.    Line 71 Chromolaena tacotana to be changed as Chromolaena tacotana (Klatt) R. M. King and H. Rob. Also include the family name as well.

4.    Line 76, Tacotanina, is it official name? Where this name comes from?

5.    Line 76, Ch. Tacotana, suddenly, this abbreviation has been brought up. The author must abbreviate in line 71, then mention this abbreviation and use it consistently for the rest of the manuscript.

6.    Section 2.1 – Before going into detail on structural characterization, the author has to briefly describe how it was isolated from Chromolaena tacotana and purified.

7.    All the decimals throughout the manuscript separated by comma (,) instead of full stop (.). For example 5,35 instead of 5.35

8.    Figure 1, adjust the numbers properly.

9.    Section 4.11 and 4.12, include the link of all the software’s used along with accessed date.

10.  The writing in the results section is great. Another issue I observed was the lack of explanation in the discussion. I would advise the authors to refocus their discussion on how the research's findings fit into the broader context of flavanones on breast cancer therapy rather than adding more background information about the literature.

11.  I advise authors to provide a critical justification for their findings and observations in the conclusion. As a result, everyone will notice the significance of this study. The conclusion must also address potential points of view. The author should stress the significance of this research component.

Author Response

Dear reviewer, 

Thanks for your time for reviewing the paper.

We sent you the answer of you questions:

It is interesting to read the manuscript “ A new Flavanone Isolated from Chromolaena tacotana (Klatt) R. M. King and H. Rob, Promotes Apoptosis in Human Breast Cancer Cells by Downregulating Antiapoptotic Proteins”. I appreciate the authors' efforts in isolating a novel flavanone and studied about its anticancer activity against human breast cancer cell lines. This will be more beneficial to scholars working drug discovery and development in breast cancer therapy using natural products. However, if the following changes are made and they are incorporated into the manuscript, it could be considered for publication in “Molecules” Journal.

Answer: Dear reviewer, thank you for the positive comments and we will develop the changes below.

  1. Avoid “Our findings/study/predicted” throughout the manuscript. Instead use “The present/current study”…I advise authors to modify the title as “A new Flavanone from Chromolaena tacotana (Klatt) R. M. King and H. Rob, Promotes Apoptosis in Human Breast Cancer Cells by Downregulating Antiapoptotic Proteins”.

Ans: "Isolated" word in the title and other words in the body text were modified according to the reviewer's recommendation. Changes were highlighted in yellow

  1. Revise the conclusion in the abstract. In the conclusion, I suggest that authors to offer a critical justification on their findings and observations. Further to be extended with one line about future direction.

Ans.:  the changes were made and highlighted in yellow

  1. Line 71 Chromolaena tacotana to be changed as Chromolaena tacotana (Klatt) R. M. King and H. Rob. Also include the family name as well.

Ans.: It was changed.

  1. Line 76, Tacotanina, is it official name? Where this name comes from?

Ans.: The name is not official, but we have given this name because it comes from the plant under study, we specify this at line 93

  1. Line 76, Ch. Tacotana, suddenly, this abbreviation has been brought up. The author must abbreviate in line 71, then mention this abbreviation and use it consistently for the rest of the manuscript.

Ans: We have changed the abbreviation appropriately in the body of the article. The changes are highlighted in yellow.

  1. Section 2.1 – Before going into detail on structural characterization, the author has to briefly describe how it was isolated from Chromolaena tacotana and purified.

We describe briefly how the flavanone was isolated from Chromolaena tacotana leaves. Line 100

  1. All the decimals throughout the manuscript separated by comma (,) instead of full stop (.). For example 5,35 instead of 5.35

Ans.: The decimals were changed

  1. Figure 1, adjust the numbers properly.

Ans.: We don't identify the wrong in the numbers in this figure, however, we organize better the  A and B position

  1. Section 4.11 and 4.12, include the link of all the software’s used along with accessed date.

Ans: Links were included as part of the references 41-43

  1. The writing in the results section is great. Another issue I observed was the lack of explanation in the discussion. I would advise the authors to refocus their discussion on how the research's findings fit into the broader context of flavanones on breast cancer therapy rather than adding more background information about the literature.

Ans: The discussion was adjusted by focusing on the activity of the flavanone, we included a possible explanation of why it could have better activity than other recognized flavones or of the same genus. We removed some paragraphs that were moved to the introduction to strengthen this section.

  1. I advise authors to provide a critical justification for their findings and observations in the conclusion. As a result, everyone will notice the significance of this study. The conclusion must also address potential points of view. The author should stress the significance of this research component.

Ans: The conclusion was adjusted according to your suggestions, thanks for this recommendation

Best regards.

Round 2

Reviewer 2 Report

All of the comments were addressed in the revised manuscript, which is now in a format that may be accepted for publication. However, full stops (.) should be used to separate all of the result numbers and decimals rather than commas (,). The entire manuscript needs to consider this modification.

Author Response

Dear Reviewer:

Thank's for your recommendation. We adjusted commas by full stops in the entire manuscript. Changes were highlighted in green.

Best Regards,